# Effect of Replacing Mineral Fertilizer with Manure on Soil Water Retention Capacity in a Semi-Arid Region

**Xiaojuan Wang** [1,2,3,4,5,*], **Lei Wang** [2] **and Tianle Wang** [2]

1   Shanxi Institute of Organic Dryland Farming, Shanxi Agricultural University, Taiyuan 030031, China
2   College of Agriculture, Shanxi Agricultural University, Taiyuan 030031, China; wl200010@163.com (L.W.); wtle1988@163.com (T.W.)
3   State Key Laboratory of Integrative Sustainable Dryland Agriculture (in Preparation), Shanxi Agricultural University, Taiyuan 030031, China
4   Key Laboratory of Sustainable Dryland Agriculture (Co-Construction by Ministry of Agriculture and Rural Affairs and Shanxi Province), Shanxi Agricultural University, Taiyuan 030031, China
5   Shanxi Province Key Laboratory of Sustainable Dryland Agriculture, Shanxi Agricultural University, Taiyuan 030031, China
*   Correspondence: wxjsxwr@163.com; Tel./Fax: +86-03517125695

**Abstract:** The long-term and excessive use of mineral fertilizers in a semi-arid region with severe water shortage will lead to soil compaction and poor water-holding capacity. The fertilization method of manure instead of mineral fertilizer has attracted wide attention. It has adverse consequences for the growth and development of crops. Hence, the objective of this study was to determine how replacing mineral fertilizer with manure affects the soil water retention curve, soil water constant, soil water availability, and soil equivalent pore size distribution, and to seek the best scheme of applying manure in semi-arid area and provide theoretical a basis for improving soil water retention capacity. Here, 0% (CK), 25% ($M_{25}$), 50% ($M_{50}$), 75% ($M_{75}$), and 100% ($M_{100}$) of 225 kg ha$^{-1}$ nitrogen from mineral fertilizer were replaced with equivalent nitrogen from manure in the Loess Plateau of China under semi-arid conditions. The centrifuge method was used to determine the soil volumetric water content under different water suction levels, and the Gardner model was used to fit and draw its soil water retention curve, and then calculate the soil water constant and equivalent pore size distribution. The results showed that the Gardner model fitted well. The soil saturated water content with the $M_{100}$ treatment was the highest, whereas the specific water capacity, water availability, and soil porosity with the $M_{75}$ treatment were the highest. The soil saturated water content showed a downward trend with the increase in nitrogen from manure instead of nitrogen from mineral fertilizer in the partial replacement treatments. This downward trend slowed down over time. The $M_{75}$ treatment increased field capacity. The $M_{100}$ treatment increased soil capillary porosity, soil available water porosity, and soil water availability compared with CK from the fifth fertilization. Replacement treatments increased the specific water capacity, soil saturated water content, soil water availability, soil porosity, and reduced the wilting point over time. In the replacement treatments, specific soil water capacity, soil water availability, and soil porosity first rose and then declined with the increase in nitrogen provided by manure replacing that provided by mineral fertilizer. Therefore, the soil water holding capacity and water supply capacity with the $M_{75}$ treatment were the best.

**Keywords:** manure; soil equivalent pore; soil water availability; soil water constant; soil water retention curve

## 1. Introduction

Soil moisture is an important factor affecting plant growth and a major driving force for the sustainable development of many terrestrial ecosystems. Changes in soil moisture have a significant impact on vegetation and soil properties [1]. Soil is the environment for crop growth and development. Soil water content affects crop yield. Too much water

in the soil will affect the respiration and nutrient loss of crop roots, and too little will affect the photosynthesis of crops. Soil water conservation improvement can increase soil water storage and organic carbon, reduce soil erosion, increase soil fertility, and increase resistance to natural disasters and pests [2]. Excessive application of mineral fertilizers in soil will lead to secondary salinization, soil pH imbalance and nutrient imbalance, and even seedling burning [3]. The soil water retention curve expresses the quantitative relationship between soil suction and soil water content [4], and it reflects the soil water holding capacity, soil water availability, and soil pore size distribution, and has important significance for the study of the storage, conservation, movement, and supply of soil moisture [5]. Soil texture and soil structure influence the soil water retention curve [6,7]. Fertilization can also affect soil structure [8]. Soil compaction was caused by the long-term application of mineral fertilizer [9], whereas the long-term application of manure improved the soil structure [10]. Therefore, to understand the variation of soil moisture in semi-arid areas [11], the combination of manure and mineral fertilizer is unalterably necessary.

Manure can promote the formation of soil aggregates and enhance soil water retention. The effects of manure combined with mineral fertilizer have been studied by many researchers [12,13]. On the basis of applying compound fertilizer 450 kg/hm$^2$ and urea 150 kg/hm$^2$, applying pig manure 4500 kg/hm$^2$, the rice yield and planting economic benefit were the highest [14]. Zhou et al. [15] observed that the effects of pig or cow manure in combination with inorganic fertilizer mainly improved the retention of capillary water but not the adsorption of hygroscopic water. Liu et al. [16] found that the combinative application of low manure rate and mineral fertilizer increased aggregate water retention capacity in the 0–10 cm layer of paddy soil. Nyamangara et al. [17] showed that the soil readily available water capacity was significantly increased by cattle manure addition. Ozlu et al. [18] observed that soil water retention was higher with manure application than with fertilizer application. In a four-year trial in the tropical sub-humid zone of the northern, Benin Mouiz W.I.A. et al. [19] found that the internal utilization efficiency increased with the increasing manure and fertilizer application.

However, little has been realized on the effect of equivalent nitrogen provided by different application rates of manure and mineral fertilizer on soil water retention curve and soil water availability. Thus, a 5-year field experiment of replacing nitrogen provided by mineral fertilizer with equivalent nitrogen provided by manure was set up. The hypothesis was that replacement treatments enhance the soil porosity, thereby increasing soil saturated water content, field capacity, water availability, and decreasing wilting point. The objectives of this study were (1) to study the effects of the replacement of nitrogen provided by mineral fertilizer with equivalent nitrogen provided by manure on soil water retention curve, soil water constant, soil water availability, and soil equivalent pore size distribution; and (2) to detect the optimum fertilization pattern to achieve higher soil water-holding capacity and higher soil water supply capacity.

## 2. Materials and Methods

### 2.1. Site Description and Experimental Design

A five-year field experiment with maize was executed in initial clay loam soil (sand 39.8%, silt 31.1%, and clay 29.1%) [16] in 2016–2020 at the Dongyang Research Station of Shanxi Agricultural University, Jinzhong, Shanxi, China (37°56′ N, 112°69′ E; 800 m altitude). Before performing the experiment, the soil sample analysis taken from the same experimental area in April 2016 showed that the top 20 cm of soil had pH 8.4 measured by $H_2O$ liquor, soil organic matter 13.0 g kg$^{-1}$, total nitrogen 1.3 g kg$^{-1}$, total phosphorus 0.9 g kg$^{-1}$, total potassium 27.1 g kg$^{-1}$, available nitrogen 51.2 mg kg$^{-1}$, available phosphorus 7.7 mg kg$^{-1}$, and available potassium 176.4 mg kg$^{-1}$. The mean annual air temperature was 11.5 °C from 2013 to 2020. The mean minimum air temperature of the coldest month (January) was −6.1 °C and the mean maximum air temperature of the hottest month (July) was 28.1 °C. The experimental site was characterized by low and unpredictable rainfall with droughts occurring at different stages of maize growth. The

long-term from 2013 to 2020 mean annual rainfall at the site was 441.2 mm, and the mean annual evaporation was 1400 mm.

The field experiment used an absolutely randomized block design with five treatments, three replicates, and a 5 m × 6 m plot. Nitrogen was provided by manure instead of 0%, 25%, 50%, 75%, and 100% of 225 kg ha$^{-1}$ nitrogen provided by mineral fertilizer in 2016–2020 (Table 1). Manure was used directly from the farm after decomposition and incorporated into approximately 0–15 cm soil depth in each experimental year in late October. The chemical composition of the manure was determined every year before application in the experiment. The 105 kg ha$^{-1}$ phosphorus provided by mineral fertilizer was applied to CK. Replacement treatments applied phosphorus provided by mineral fertilizer with the 105 kg ha$^{-1}$ minus the phosphorus content of manure incorporated into soil. The mineral nitrogen and phosphorus fertilizers were applied severally as basal fertilizers before sowing maize. Urea (N 46%), monoammonium phosphate (N 12%, $P_2O_5$ 61%), and sheep manure were used. The same treatment was applied to the same experimental plot every year. The total nitrogen content of the organic fertilizer applied in the autumn of 2018 was 20.43 g/kg, and the total phosphorus content was 9.17 g/kg; the total nitrogen content of organic fertilizer applied in autumn 2019 was 23.97 g/kg, and the total phosphorus content was 10.97 g/kg. In each experimental year, the *Dafeng 30* maize variety was cultivated at a rate of 49,500 plants ha$^{-1}$ in late April or early May and harvested in late September. Maize was grown every year of experiment without crop rotation. Maize yield was determined by harvesting 20 ears of maize in the middle of each plot. Due to the insignificant changes in soil moisture characteristics from 2016 to 2018, we showed the analysis from 2019 to 2020.

**Table 1.** Different replacement treatments in 2016–2020.

| Treatments | 225 kg ha$^{-1}$ Nitrogen from Fertilizer | 225 kg ha$^{-1}$ Nitrogen from Manure |
|:---:|:---:|:---:|
| CK | 100% | 0% |
| ($M_{25}$) | 75% | 25% |
| ($M_{50}$) | 50% | 50% |
| ($M_{75}$) | 25% | 75% |
| ($M_{100}$) | 0% | 100% |

*2.2. Sampling and Analysis Methods*

Soil samples used for measuring soil water retention curve were collected using a 100 cm$^3$ cutting ring at the plow layer after maize harvest. The soil samples were saturated slowly (>24 h), and then weighed, finally put into a CR22N high-speed constant temperature centrifuge (Japan-Hitachi) starting from full saturation at 20 °C. Use weights to adjust the weight of the four centrifuge tanks to the same level, then close the centrifuge cover, set the speed and time, and press the Start button to start the centrifuge. After centrifugation, soil sample weight was measured at 10, 30, 50, 80, 100, 300, 500, 800, 1000, and 1500 kPa. Subsequently, the soil samples were oven-dried at 105 °C for 24 h following weighing. The volumetric water content at different suction levels and soil saturated water content at 0 kPa were calculated using the equation:

$$\theta = \frac{V_W}{V} = \frac{W_S - W_o}{\rho \times V} \tag{1}$$

where $\theta$ is the soil volumetric water content at a certain suction (cm$^3$ cm$^{-3}$); $V_W$ is the volume of water of soil sample at a certain suction (cm$^3$); $V$ is the volume of soil sample with 100 cm$^3$ (cm$^3$); $W_S$ is the soil sample weight at a certain suction (g); $W_o$ is the soil sample weight after oven-drying (g); and $\rho$ is the water density with 1 g cm$^{-3}$ (g cm$^{-3}$) [20].

The Gardner model was used to fit the acquired data using Excel 2016 as follows:

$$\theta = A \times S^{-B} \tag{2}$$

where $\theta$ is soil volumetric water content (cm$^3$ cm$^{-3}$); $A$ and $B$ are dimensionless parameters related to the curve shape; and $S$ is soil water suction (kPa). Parameter A determined the height of the curve and level of water-holding capacity. The larger the value of A, the stronger the water-holding capacity. Parameter B determined how fast the soil water content decreased with the increase in soil water suction. The larger the value of B, the greater the curvature of the curve and the more intense the change [21].

Specific soil water capacity was derived from Formula (2). It was defined as

$$C = A \times B \times S^{-(B+1)} \tag{3}$$

where $C$ is specific soil water capacity (kPa$^{-1}$); $A$ and $B$ are dimensionless parameters related to the curve shape; and $S$ is soil water suction (kPa) [22].

The field capacity, soil volumetric water content at 600 kPa, and wilting point were calculated by the Gardner model at 33,600 and 1500 kPa, respectively [23]. Readily available water content, delayed available water content, and available water content were defined as

$$\theta_r = \theta_f - \theta_{600} \tag{4}$$

$$\theta_d = \theta_{600} - \theta_w \tag{5}$$

$$\theta_a = \theta_f - \theta_w \tag{6}$$

where $\theta_r$ is the readily available water content (cm$^3$ cm$^{-3}$); $\theta_f$ (cm$^3$ cm$^{-3}$), the field capacity; $\theta_{600}$ (cm$^3$ cm$^{-3}$), the soil volumetric water content at 600 kPa; $\theta_d$ (cm$^3$ cm$^{-3}$), the delayed available water content; $\theta_w$ (cm$^3$ cm$^{-3}$), the wilting point; and $\theta_a$ (cm$^3$ cm$^{-3}$) is the available water content [23].

The pore size of capillary porosity was 0.03–0.1 mm, while the pore size of available water porosity was 0.002–0.06 mm [20]. Water suction of capillary porosity and available water porosity was 3–10 kPa and 5–150 kPa, respectively [23]. The soil volumetric water content at 3, 5, 10, and 150 kPa was calculated using Formula (2). The capillary porosity was the soil volumetric water content at 3 kPa minus the soil volumetric water content at 10 kPa multiplied by 100%. The available water porosity was the soil volumetric water content at 5 kPa minus the soil volumetric water content at 150 kPa multiplied by 100% [23].

*2.3. Statistical Analysis*

Analysis of variance (ANOVA) was performed using SAS 6.2 for Windows 8. The significance of treatment effects was determined using the F-test. Multiple comparisons of means were performed using Duncan's multiple range test at the $p \leq 0.05$ level.

**3. Results**

*3.1. Soil Water Retention Curve*

Figure 1 shows the soil water retention curve of each treatment. The points are the measured values, whereas the lines are the fitted values in Figure 1. The curve significantly changed when water suction was 100 kPa (Figure 1). The soil water content of each treatment showed a rapid decline trend when water suction was lower than 100 kPa, whereas the soil water content of each treatment showed a slow decline trend when the water suction was greater than 100 kPa.

To quantitatively study soil water retention curve, the Gardner model was used to fit the measured data of the soil water retention curve. The fitting coefficient R$^2$ of each treatment was above 0.98 (Table 2). Thus, the fitting effect was good. In 2019, parameter A with the M$_{75}$ treatment was greater than that with CK, whereas parameter A with all other replacement treatments was lower than that with CK, indicating that the soil water holding capacity of the M$_{75}$ treatment was better than CK, whereas the soil water storage capacity of the other replacement treatments was not as good as that of CK. In 2020, parameter A

of each treatment was greater than or equal to CK, indicating that the soil water holding capacity increased over time with replacement treatments. In the replacement treatments, the value of parameter A with the M$_{75}$ treatment was the largest; parameter A showed a trend of first increasing and then decreasing with the increase in nitrogen from manure instead of nitrogen from mineral fertilizer, indicating that the M$_{75}$ treatment had the best soil water holding capacity. The soil water holding capacity rose first and then declined with the increase in nitrogen from manure instead of nitrogen from mineral fertilizer. Parameter B with replacement treatments was greater than that with CK, indicating that soil water content with replacement treatments decreased faster with the increase in soil water suction compared with CK.

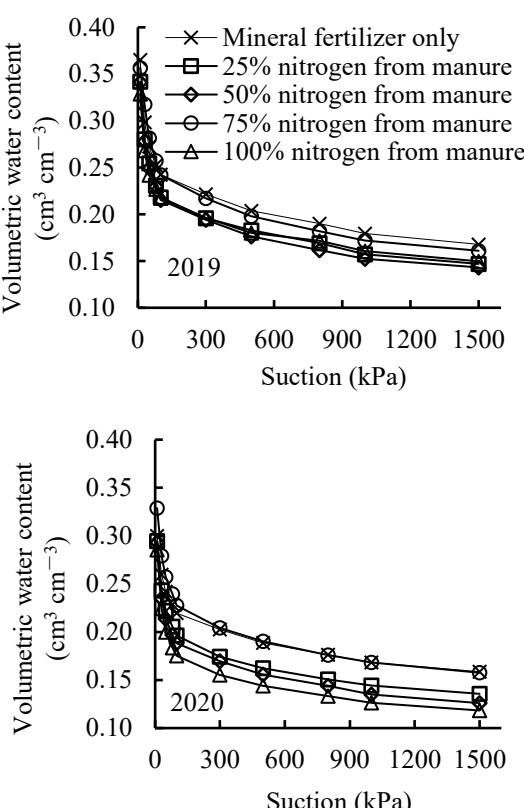

**Figure 1.** Soil water retention curve as a function of the different replacement treatments in 2019–2020.

**Table 2.** Parameters in the modeling of the soil water retention curve as a function of the different replacement treatments in 2019 and 2020.

| Years | Treatments | *A* | *B* | *R*$^2$ |
|---|---|---|---|---|
| 2019 | Mineral fertilizer only | 0.490 | 0.145 | 0.99 |
| | 25% nitrogen from manure | 0.479 | 0.160 | 0.99 |
| | 50% nitrogen from manure | 0.486 | 0.166 | 0.99 |
| | 75% nitrogen from manure | 0.524 | 0.159 | 0.99 |
| | 100% nitrogen from manure | 0.441 | 0.146 | 0.99 |
| 2020 | Mineral fertilizer only | 0.393 | 0.121 | 0.99 |
| | 25% nitrogen from manure | 0.402 | 0.149 | 0.99 |
| | 50% nitrogen from manure | 0.405 | 0.158 | 0.99 |
| | 75% nitrogen from manure | 0.451 | 0.142 | 1.00 |
| | 100% nitrogen from manure | 0.393 | 0.165 | 0.98 |

Notes: *A* and *B* are dimensionless parameters related to the curve shape. *R*$^2$ is coefficient of determination.

### 3.2. Specific Soil Water Capacity

Specific soil water capacity reflects the water content variation in soil caused by unit suction change and is an important index to evaluate soil water supply capacity at

different suction levels [21]. The value of specific soil water capacity can reflect soil water supply capacity well at soil water suction of 100 kPa. The larger the value of specific soil water capacity, the better the soil water supply capacity [24]. Values of specific soil water capacity with each treatment at 100 kPa are shown in Figure 2. Replacement treatments had higher specific water capacity values compared with CK except the $M_{100}$ treatment in 2019. However, the specific soil water capacity value of each replacement treatment was higher than that of CK in 2020. Thus, the soil water supply capacity with replacement treatments increased over time. The $M_{75}$ treatment had the largest specific soil water capacity value among the replacement treatments. The specific soil water capacity of the $M_{75}$ treatment was 9.89% and 22.43% ($p \leq 0.05$) higher than that of CK in 2019 and 2020, respectively. Therefore, the $M_{75}$ treatment had the strongest soil water supply capacity and the highest drought resistance capacity. Specific soil water capacity rose first and then declined with the increase in nitrogen from manure instead of nitrogen from mineral fertilizer, indicating that the soil water supply capacity rose first and then declined with the increase in nitrogen from manure instead of nitrogen from mineral fertilizer.

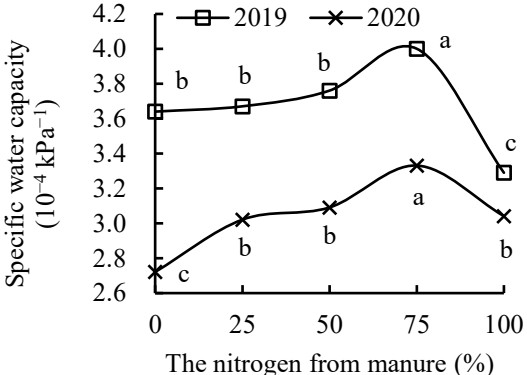

**Figure 2.** Specific soil water capacity at 100 kPa as a function of the different replacement treatments in 2019–2020. Note: The different letters in the same year in the graph indicate significant differences ($p \leq 0.05$).

### 3.3. Soil Water Constant

The soil saturated water content of the $M_{100}$ treatment was slightly higher than that of CK, and that of all other replacement treatments was less than CK in 2019 (Figure 3). The soil saturated water content of the $M_{100}$ treatment was the highest, and 16.9% ($p \leq 0.05$) higher than that of CK; that of the $M_{25}$, $M_{50}$, and $M_{75}$ treatments was slightly higher than that of CK ($p > 0.05$) in 2020. Thus, complete substitution of the nitrogen provided by mineral fertilizer with the nitrogen provided by manure increased the soil saturated water content and enhanced the maximum soil water holding capacity over time. The soil saturated water content showed a downward trend with the increase in nitrogen from manure instead of nitrogen from mineral fertilizer in the partial replacement treatments. The downward trend was slower in 2020 than in 2019.

The changing trend of field capacity was the same for the two years. With the exception of the field capacity of the $M_{75}$ treatment, which was higher than CK, the field capacity of the other replacement treatments was lower than CK (Figure 4). The wilting point of all replacement treatments was lower than that of CK (Figure 5). With the increase in nitrogen from manure instead of nitrogen from mineral fertilizer, both field capacity and wilting point showed a trend of decline, then rise, and again decline.

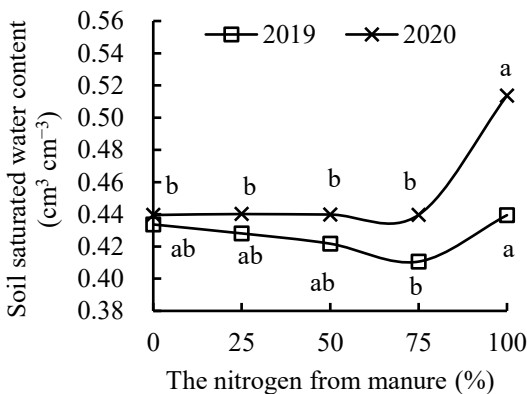

**Figure 3.** Saturated water content as a function of the different replacement treatments in 2019–2020. Note: The different letters in the same year in the graph indicate significant differences ($p \leq 0.05$).

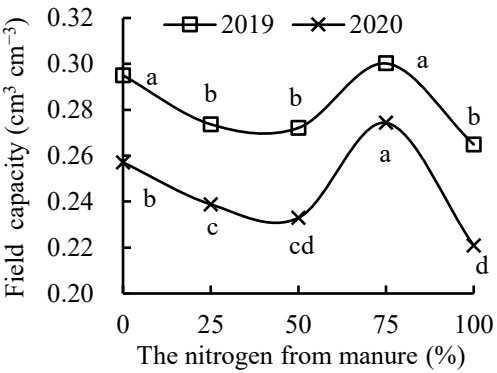

**Figure 4.** Field capacity as a function of the different replacement treatments in 2019–2020. Note: The different letters in the same year in the graph indicate significant differences ($p \leq 0.05$).

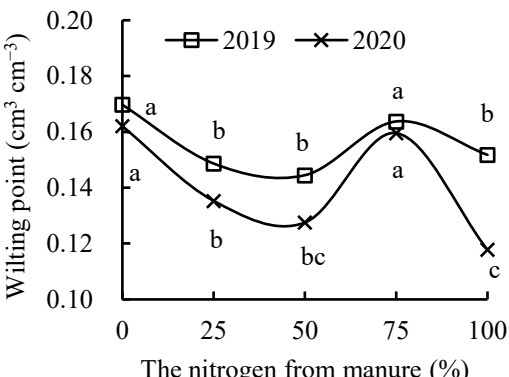

**Figure 5.** Wilting point as a function of the different replacement treatments in 2019–2020. Note: The different letters in the same year in the graph indicate significant differences ($p \leq 0.05$).

Replacement treatments increased the readily available water content compared with CK ($p \leq 0.05$), except for the $M_{100}$ treatment in 2019 (Figure 6). The readily available water content with $M_{25}$, $M_{50}$, $M_{75}$, and $M_{100}$ treatments was 10.12%, 12.52%, 21.70%, and 21.68% ($p \leq 0.05$) higher than that of CK, respectively, in 2020. Thus, replacing nitrogen from mineral fertilizer with nitrogen from manure could increase the readily available water content over time. The readily available water content with the $M_{75}$ treatment was the highest, and 9.48% and 21.68% ($p \leq 0.05$) higher than that with CK in 2019 and 2020, respectively. The readily available water content rose first and then declined with the increase in nitrogen from manure instead of nitrogen from mineral fertilizer.

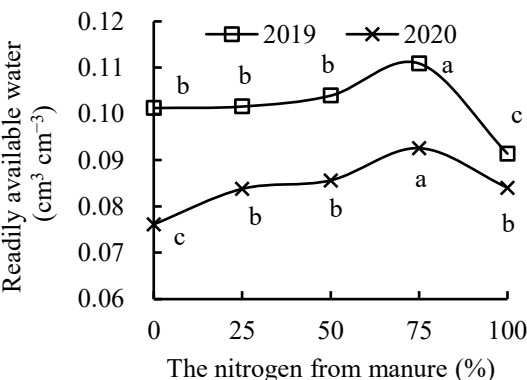

**Figure 6.** Readily available water content as a function of the different replacement treatments in 2019–2020. Note: The different letters in the same year in the graph indicate significant differences ($p \leq 0.05$).

The delayed available water content with the $M_{75}$ treatment was higher than that with CK, and that with the other replacement treatments was lower than CK in 2019 (Figure 7). $M_{75}$ treatment significantly increased the delayed available water content ($p \leq 0.05$), whereas other replacement treatments slightly increased it ($p > 0.05$) compared with CK in 2020. These indicated that the replacement of nitrogen from mineral fertilizer with nitrogen from manure could increase the delayed available water content over time. The delayed available water content with the $M_{75}$ treatment was the highest, and 6.64% and 16.84% ($p \leq 0.05$) higher than that with CK in 2019 and 2020, respectively. In the replacement treatments, the delayed available water content showed a trend of first increasing and then decreasing as the amount of nitrogen from manure substituted for nitrogen from mineral fertilizer increased.

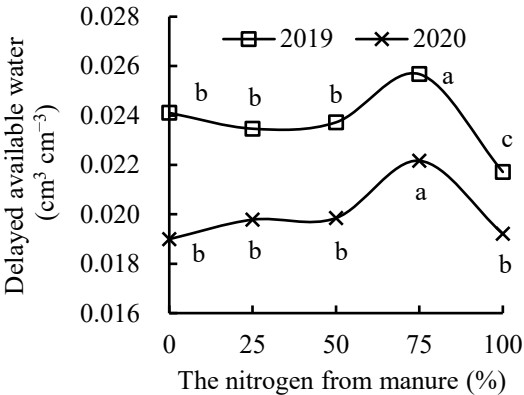

**Figure 7.** Delayed available water content as a function of the different replacement treatments in 2019–2020. Note: The different letters in the same year in the graph indicate significant differences ($p \leq 0.05$).

Available water content of the $M_{50}$ and $M_{75}$ treatments was higher than that of CK, and that of all other replacement treatments was lower than that of CK in 2019 (Figure 8). Replacement treatments increased the available water content by 8.52–20.72% ($p \leq 0.05$) compared with CK in 2020. These indicated that the substitution of nitrogen from manure for nitrogen from mineral fertilizer could increase the available water content over time. The available water content with the $M_{75}$ treatment was the highest, and 8.93% and 20.72% ($p \leq 0.05$) higher in 2019 and 2020, respectively, compared with CK. In the replacement treatments, the available water content rose first and then declined with the increase in nitrogen from manure instead of nitrogen from mineral fertilizer.

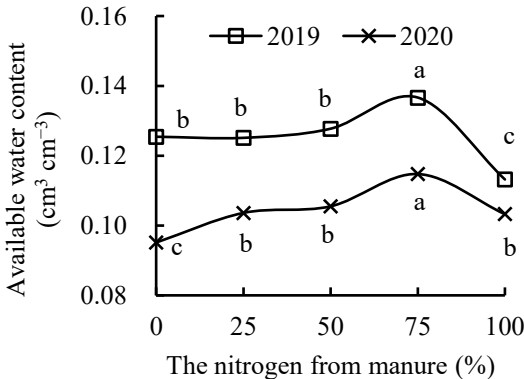

**Figure 8.** Available water content as a function of the different replacement treatments in 2019–2020. Note: The different letters in the same year in the graph indicate significant differences ($p \leq 0.05$).

### 3.4. Soil Equivalent Pore Size Distribution

Replacement treatments increased the soil capillary porosity by 5.23–14.50% ($p \leq 0.05$), except for the $M_{100}$ treatment compared with CK in 2019, whereas the replacement treatments increased soil capillary porosity by 20.39–30.04% ($p \leq 0.05$) compared with CK in 2020 (Figure 9). These indicated that the replacement treatments could increase soil capillary porosity, facilitate soil water conduction, reduce water accumulation in the soil interior and surface, retain soil moisture, and ensure crops grow well over time [23].

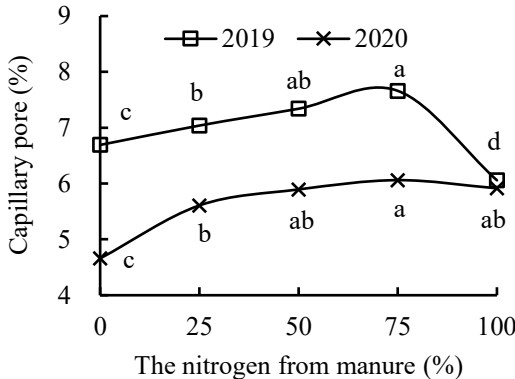

**Figure 9.** Capillary porosity of soil as a function of the different replacement treatments in 2019–2020. Note: The different letters in the same year in the graph indicate significant differences ($p \leq 0.05$).

The soil available water porosity of replacement treatments was higher than that of CK, except for the $M_{100}$ treatment in 2019 (Figure 10). Replacement treatments increased the soil available water porosity by 15.41–26.06% ($p \leq 0.05$) compared with CK in 2020. These indicated that the replacement treatments could gradually increase soil available water porosity and was conducive to soil water permeability, soil water retention, and the formation of a good soil structure [23].

The $M_{75}$ treatment had the highest soil capillary porosity and soil available water porosity. With the increase in nitrogen from manure instead of nitrogen from mineral fertilizer, the soil capillary porosity and soil available water porosity rose first and then declined.

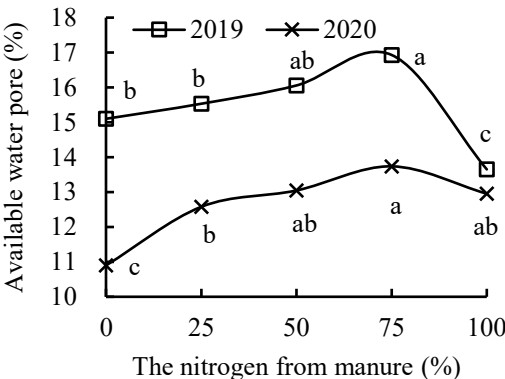

**Figure 10.** Available water porosity of soil as a function of the different replacement treatments in 2019–2020. Note: The different letters in the same year in the graph indicate significant differences ($p \leq 0.05$).

## 4. Discussion

The soil water storage with water suction can be determined by the soil water retention curve [25]. Wang et al. [24] reported that soil moisture rapidly declined at 0–6 kPa water suction, decreased slowly at 6–100 kPa water suction, and was stable at 100–1500 kPa water suction in the long-term application of organic–inorganic fertilizer in lateritic soil. The soil water retention curve in the present study indicated that soil water retention capacity decreased rapidly first and then slowly, with the critical point being the water suction level 100 kPa. This might be because both macropores and micropores retained soil water when water suction was lower than 100 kPa [23], whereas only micropores could retain part of water when the water suction was greater than 100 kPa [21]. When the NaCl concentration was 3%, the small pores, micropores, and extremely micropores of the soil increased, and the pores and macropores decreased [26]. This study shows that the $M_{100}$ treatment increased the soil saturated water content, whereas the other treatments decreased it. This might be due to the higher content of organic matter in $M_{100}$ treatment. In contrast, chemical fertilizers are fast-acting nutrients. They can quickly provide the necessary nutrients for plant growth, promote root development, and increase soil porosity.

Chakraborty et al. [27] suggests that the field capacity was significantly higher in farmyard manure plots compared with no fertilizer or manure. Fouladidorhani et al. [28] revealed that manure combined with biochar increased water content at field capacity in contrast with no fertilizer in saline–sodic silt loam soil. The combined application of organic and inorganic fertilizers can increase the stability of soil aggregates in fluvo-aquic soil [29]. Studies have shown that the proportion of a small amount of manure and a large amount of N fertilizer may accelerate the decomposition of organic matter, thus affecting the agglomeration effect [30]. In this study, compared with single application of mineral fertilizer, the $M_{75}$ treatment had higher field capacity, whereas the other replacement treatments had lower field capacity. This might be because of the higher soil porosity with the $M_{75}$ treatment (Figures 9 and 10).

This study showed that the replacement treatments decreased the wilting point compared with single application of mineral fertilizer. Thus, even when the soil water content was still high, the soil water potential of single application of mineral fertilizer was close to the wilting point and could not provide sufficient water to crops. This study showed that with the increase in nitrogen from manure instead of nitrogen from mineral fertilizer, both field capacity and wilting point showed a trend of decline, then rise, and again decline. The reasons for this phenomenon require further research.

Wang et al. [24] reported that long-term organic–inorganic fertilizer increased the soil saturated water content of red soil. Du et al. [31] found that the application of manure could effectively improve soil fertility, increase the survival rate of watermelon, and thus increase the yield of watermelon. Zhou et al. [32] showed that plant available water content was not significantly improved by pig or cow manure in combination with inorganic fertilizer.

Feng et al. [33] illustrated that poultry litter addition significantly increased plant available water content by 20%. Singh et al. [34] indicated that long-term application of cattle manure and fertilizer increased soil porosity compared with no fertilizer at the microscale level in maize–soybean rotation. Xu et al. [35] suggested that the total macroporosity (>50 μm) increased from 7.95% to 16.36% throughout the whole organic plantation (one-year-old, nine-year-old, and fourteen-year-old fields), and a similar trend occurred at small (50–500 μm) and medium (500–1000 μm) macroporosity. In this study, replacement treatments increased the soil saturated water content, soil water availability, soil capillary porosity, and soil available water porosity compared with single application of mineral fertilizer over time. This might be because the ammonium ion of mineral fertilizer was absorbed by plants; meanwhile, acidic soil was formed by the rest of the acid radicals of the mineral fertilizer combined with hydrogen ion of soil after long-term application of acidic fertilizers (monoammonium phosphate) in alkaline soil, eventually resulting in hardened soil, increased pH, and soil structure destruction [36]. In this study, M100 treatment had the lowest readily available water content in 2019. Meanwhile, in the replacement treatments, the available water content rose first and then declined with the increase in nitrogen from manure instead of nitrogen from mineral fertilizer. These may be due to the slower release of nutrients when applying manure.

Except for the soil saturated water content, other soil water properties were higher in 2019 compared to 2020, which might be related to the uneven rainfall distribution throughout the year.

Replacing the nitrogen from chemical fertilizer with the nitrogen from organic fertilizer could have positive effects on soil water properties, thus playing a promoting role in agricultural practices in semi-arid regions. At the same time, it also contributed to the efficient utilization of water resources and the achievement of sustainable agriculture.

## 5. Conclusions

The substitution of nitrogen from manure for nitrogen from mineral fertilizer gradually increased the soil water holding capacity compared with applying mineral fertilizer alone. A single application of manure enhanced soil porosity and soil water availability compared with a single application of mineral fertilizer from the fifth fertilization year at the same nitrogen rate. A fertilization combination consisting of 75% nitrogen provided by manure and 25% nitrogen provided by mineral fertilizer was found to be a suitable approach to achieve optimal effects of soil water retention in semi-arid regions, taking into consideration factors such as the soil water retention curve, soil water availability, and soil porosity.

**Author Contributions:** X.W.: Investigation, Formal analysis, Writing—Original draft, Funding acquisition; L.W.: Writing—Revised draft, Editing; T.W.: Investigation. All authors have read and agreed to the published version of the manuscript.

**Funding:** This work was supported by the youth top-notch talent support program of Shanxi province (grant number HNZXBJ001); State Key Laboratory of Sustainable Dryland Agriculture, Shanxi Agricultural University (Grant number 202105D121008-1-7); and the Special Fund for Agro-scientific Research in the Public Interest (grant number 201503124).

**Data Availability Statement:** The data presented in this study are available on request from the corresponding author. The data are not publicly available due to privacy.

**Conflicts of Interest:** The authors declare no conflict of interest.

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
