# Peer review of "Effect of Replacing Mineral Fertilizer with Manure on Soil Water Retention Capacity in a Semi-Arid Region"

_agronomy, doi:10.3390/agronomy13092272_

Round 1

Reviewer 1 Report (New Reviewer)

Dear authors,

After the review, I think that your work needs more substantial editing. The idea of the research is OK, but there are some shortcomings that should be corrected or supplemented.

After you have corrected it, I will make sure that your work is ready for publication.

Author Response

Response to Reviewer 1 Comments

Point 1: Lines 73 - 85  I believe that the paragraph should be standardized - first write something about the climatic parameters.of the soil (physical and chemical properties - clearly state that the properties shown in this paragraph are the initial state of the soil - that is, before performing the experiment), and then about the climatic and meteorological data of the researched area - specify the period ( perennial) that you refer to in the text (xxxx-xxxx).

Response 1: Thanks for your careful review. I have changed the location of soil properties and meteorological parameters to make the data more compact, and also added the corresponding time of meteorological parameters.

Point 2: Lines 79  I believe that it should be stated in which solvent the pH was determined (KCl, H20 or CaCl).

Response 2: Added reagent solution to test PH : H2O.

Point 3: Lines 86-97  I think it would be better to present this paragraph in a table, because it is quite confusing this way. I think it would be more clear and understandable in a table.

Response 3: Thanks for your careful review. I 've replaced grouped text data with tables.

Point 4: Lines 97 - 105   I believe that the chemical composition of the manure used in the experiment should be stated.

It is also necessary to provide more information about manure:

  1. Was it pelletized or was manure used directly from the farm.
  2. Was the chemical composition of the manure determined every year before application in the experiment?

It should be better explained if the same treatment was applied to the same experimental sex every year

Was maize grown every year of experiment? without crop rotation?

Was the maize yield determined on each experimental plot (each treatment)?

Response 4: First of all, manure is directly used from the farm, and it is measured before the application of fertilizer every year. The same treatment was adopted in the same experiment, and the chemical composition of fertilizer and manure and the yield of corn have also been explained. In addition, corn does not rotate.

Point 5: Lines 108  please indicate the volume of the rings.

Response 5: Thanks for your careful review. I have indicated the volume of the ring.

Point 6: Lines 147 - 151  Nowhere in the text (in the Results chapter) are the P values obtained after statistical analysis stated. Please provide that information.

Response 6: I added information about P in the results section.

Point 7: Generaly for Results  Why are the results presented only for 2019 and 2020, and the research lasted 5 years? For the other years, the analysis was not carried out? If so, then it should be clearly stated in the Materials and Methods (the experiment was conducted over 5 years, but the results were processed only for 2019 and 2020 years).

In the Materials and Method chapter, the results of the analysis of the basic properties of the soil before the start of the experiment are presented, but the results of the initial state of the soil with regard to retention curves, porosity, specific water capacity, soil saturated water content, field capacity wilting point and so on are not presented.

I believe that the stated results should also be shown, because the title of the paper is related to changes in soil water capacity due to changes in nutrition (mineral fertilizer - manure).

Response 7: Due to the insignificant changes in soil moisture characteristics from 2016 to 2018, we showed the analysis from 2019 to 2020. We are sorry that during the course of the experiment, we did not measure data on soil moisture characteristic.The reason why the survey parameters are higher in 2019 than in 2020 is also explained in the discussion.

Point 8: Lines 215 - 220  How do you explain this, Say a few words to this in the discussion section

Response 8:  It explains why soil saturated water content and maximum water holding capacity increase with the increase of manure instead of mineral fertilizer over time.This explanation is discussed in Lines 378-391 of the discussion section of the revised draft.

Point 9: Figure 3.  How do you explain the drop in Saturated water content with the increase in added manure (except M100 treatment)

Response 9: The interpretation of Figure 3 has been revised to indicate the P value. It is clearly pointed out that the nitrogen provided by mineral fertilizer is completely replaced by the nitrogen provided by manure, which increases the saturated water content of the soil and enhances the maximum water holding capacity of the soil.

Point 10: Lines 226 - 228  How do you explain this.Say a few words to this in the discussion section

Response 10: In the discussion, it was mentioned that the field capacity and wilting point showed a trend of decreasing first, then increasing and then decreasing, but the reasons for this phenomenon need to be further studied.

Point 11: Lines 234 -235  Are these values significantly higher? missing P values from statistics?

10.12% specify for which treatment?

21.68% specify for which treatment?

Response 11:  I have added the missing P value. M25, M50, M75 and M100 treatments was 10.12%, 12.52%, 21.70% and21.68%.

Point 12: Figure 6.  How do you explain that the M100 treatment has the lowest values?

Response 12: The interpretation of Figure 6 has been corrected and the P value has been marked.

Point 13: Lines 246 - 253  Are the values statistically higher? P-values?

how do you explain part of the paragraph from the Lines 250 - 253.

Response 13: The interpretation of Figure 7 has been corrected and the P value has been marked. In the 280 Liness of the revised draft, it was explained that M75 was significantly higher than other treatment groups, so the delayed effective water content increased first and then decreased.

Point 14: Drought stress was not done, so this should be thrown out because it is not possible to comment on things that were not primarily done in the experiment

Response 14: Thanks for your careful review. Articles on the discussion of drought stress have been deleted from the article.

Reviewer 2 Report (New Reviewer)

The paper discusses the study's focus on the impact of replacing mineral fertilizer with manure on soil water retention capacity in a semi-arid region. The strengths of the paper include its clear research objective and methodology, which involves various replacement proportions of nitrogen from manure in the Loess Plateau of China. The use of the centrifuge method and the Gardner model for data analysis is well-explained. The paper highlights key findings, such as the relationship between replacement treatments and soil water content, capacity, porosity, and availability.

  1. The level of English throughout the Manuscript needs to meet the journal's standard. Therefore, you may wish to ask a native speaker to check your Manuscript for grammar, style, and syntax.
  2. The References should be prepared according to Agronomy journal standards.
  3. Many old references are mentioned; if not, standards replace them with new and recent ones. Please update from 2015-2023.
  4. The abstract must have a rationale, an objective, materials and methods, results, and conclusions. The first sentence must be a rationale and research Gap for this study.
  5. The introduction should be improved by inserting new information from recent studies about the impact of replacing mineral fertilizer with manure on soil water retention capacity in a semi-arid region
  6. The Introduction begins abruptly without providing context or background information on the importance of soil water retention and the challenges of excessive mineral fertilizer use. Consider introducing the broader significance of the study's topic to engage readers.
  7. For Materials and Methods: While the centrifuge method and Gardner model are mentioned, more details on their application could enhance the M and M clarity. Describe how the centrifuge method measures soil volumetric water content and how the Gardner model helps analyze and interpret soil water retention data.
  8. The paper lacks a discussion of the implications of the findings. Address how the observed changes in soil water properties due to manure replacement could benefit agricultural practices, water conservation, or sustainability in semi-arid regions. In addition, The paper doesn't compare the effects of manure replacement treatments with conventional mineral fertilizer applications. Including a brief comparison could highlight the advantages of the proposed approach.
  9. The discussion must be substantially improved. The authors mostly only make comparisons of their results with the literature's results. However, they need to discuss the mechanisms by which the results are obtained. 

he level of English throughout the Manuscript needs to meet the journal's standard. Therefore, you may wish to ask a native speaker to check your Manuscript for grammar, style, and syntax

Author Response

Response to Reviewer 2 Comments

Point 1: The level of English throughout the Manuscript needs to meet the journal's standard. Therefore, you may wish to ask a native speaker to check your Manuscript for grammar, style, and syntax.

Response 1: The manuscript had been revised by TopEdit (www.topeditsci.com) for linguistic assistance during the preparation of this manuscript.

Point 2: The References should be prepared according to Agronomy journal standards.

Response 2: Thanks for your careful review.I have revised the references according to the standard of Agronomy journals.

Point 3: Many old references are mentioned; if not, standards replace them with new and recent ones. Please update from 2015-2023.

Response 3: Thanks for your careful review. I have updated the references and deleted too old citations.

Point 4: The abstract must have a rationale, an objective, materials and methods, results, and conclusions. The first sentence must be a rationale and research Gap for this study.

Response 4: The abstract is revised. The first sentence is the theoretical basis and research gap of this study. The abstract also includes a principle, purpose, material and method, and conclusion.

Point 5: The introduction should be improved by inserting new information from recent studies about the impact of replacing mineral fertilizer with manure on soil water retention capacity in a semi-arid region.

Response 5: The introduction includes the latest research on the effect of replacing mineral fertilizers with manure on soil water retention capacity in semi-arid areas.

Point 6: The Introduction begins abruptly without providing context or background information on the importance of soil water retention and the challenges of excessive mineral fertilizer use. Consider introducing the broader significance of the study's topic to engage readers.

Response 6: At the beginning of the introduction, the importance of soil water conservation and the harm of excessive application of mineral fertilizers were added to enhance the transitional nature of the article.

Point 7: For Materials and Methods:While the centrifuge method and Gardner model are mentioned, more details on their application could enhance the M and M clarity. Describe how the centrifuge method measures soil volumetric water content and how the Gardner model helps analyze and interpret soil water retention data.

Response 7: In Lines 129-132 of the revised draft, we carefully described how the centrifugal method measured soil volumetric water content, and Lines 145-151 explained how the Gardner model helped analyze and interpret soil water retention data.

Point 8: The paper lacks a discussion of the implications of the findings. Address how the observed changes in soil water properties due to manure replacement could benefit agricultural practices, water conservation, or sustainability in semi-arid regions. In addition, The paper doesn't compare the effects of manure replacement treatments with conventional mineral fertilizer applications. Including a brief comparison could highlight the advantages of the proposed approach.

The discussion must be substantially improved. The authors mostly only make comparisons of their results with the literature's results. However, they need to discuss the mechanisms by which the results are obtained.

Response 8: In the discussion, I added the comparison results of soil saturated water content, the change trend of field capacity and wilting point, and the possible reasons for the results. It explains why soil saturated water content and maximum water holding capacity increase with the increase of manure instead of mineral fertilizer over time. The discussion has been substantially modified.

Reviewer 3 Report (Previous Reviewer 2)

The quality of the article is low and does not meet the standard of the Agronomy journal. The description of the experiment is chaotic and imprecise. The visual quality of the figures is unacceptable. Even the conclusion of the article is very weak. The article must bring something new.

It would help if you tried to publish in another journal (or completely rework the article and significantly improve it).

Author Response

Response to Reviewer 3 Comments

Thank you especially for your comments, I made a big change to the article. The manuscript had been revised by TopEdit (www.topeditsci.com) for linguistic assistance during the preparation of this manuscript.

First of all, I integrated the scattered data, added the specific year of meteorological parameters, The abstract also includes a principle, purpose, material and method, and conclusion. At the beginning of the introduction, the importance of soil water conservation and the harm of excessive application of mineral fertilizers were added to enhance the transitional nature of the article.The introduction includes the latest research on the effect of replacing mineral fertilizers with manure on soil water retention capacity in semi-arid areas.I have revised the references according to the standard of Agronomy journals and updated the references and deleted too old citations.

Then, a more in-depth and intuitive description of the experiment was made by replacing the tedious text in Lines 89-97 with tables. In Lines 129-132 of the revised draft, we carefully described how the centrifugal method measured soil volumetric water content, and Lines 145-151 explained how the Gardner model helped analyze and interpret soil water retention data. The picture was modified to increase the P value, indicating that the difference was significant.

Finally, I made substantial modifications to the results. in the discussion, I added the comparison results of soil saturated water content, the change trend of field capacity and wilting point, and the possible reasons for the results. It explains why soil saturated water content and maximum water holding capacity increase with the increase of manure instead of mineral fertilizer over time. The discussion has been substantially modified.

Round 2

Reviewer 2 Report (New Reviewer)

The References should be prepared according to Agronomy journal standards in the text.

Author Response

Response to Reviewer 2 Comments

Point 1: The References should be prepared according to Agronomy journal standards.

Response 1: Thanks for your careful review. I have revised the references according to the standard of Agronomy journals.

Reviewer 3 Report (Previous Reviewer 2)

The article has been improved. But not enough. Certain quality standards must be maintained because the Agronomy journal falls under the Q1 category. I recommend adding at least another ten cited sources - especially to the Discussion chapter. Also, improve the Conclusion.

Author Response

Response to Reviewer 3 Comments

Point 1: I recommend adding at least another ten cited sources - especially to the Discussion chapter. Also, improve the Conclusion.

Response 1: Thanks for your careful review. I have added the corresponding references, and further improved the conclusion of the article.

This manuscript is a resubmission of an earlier submission. The following is a list of the peer review reports and author responses from that submission.

Round 1

Reviewer 1 Report

Manuscript ID: agronomy-2441095

Comments and Suggestions for Authors as follows:

Title: Effects of replacing mineral fertilizer with straw incorporation on soil water-holding characteristics

-The addition of straw or plant residues to the soil directly without decomposition outside the soil results in a significant decrease in the soil nitrogen content.

- In addition to a decrease in temperature to very large rates in the period following the addition of corn straw until corn cultivation.

- The introduction is very short and does not cover parts of the study.

- The authors stated that the study was conducted over five years, while results for only three years were presented?.

- In the results section (line 179), the authors mentioned that the field capacity increases at S25, while it decreases with the rest of the treatments that contain an increase in the rate of maize straw, while it is assumed that the opposite is true as was done in previous studies (lines 293-295).

- There is no discussion of the results other than what has been reviewed for some previous studies and their results.

- The study did not address the replacement of mineral nitrogen with nitrogen from maize straw, either in the results section or the discussion section.

- References are very few and very short, as they did not cover all points of the study.

- The authors did not follow the system used in writing references and citations according to the Journal : Agronomy.

Abstract

Line 65: (Liu et al., 2017) : What does this reference refer to? 

Reviewer 2 Report

The article contains more or less predictable results. Therefore, it should be written as well as possible to have a chance to succeed. Unfortunately, essential parts, such as the Discussion, are at a low level. Also, the total number of references (24) needs to be increased. It is necessary to cite the latest world studies.
